# Enhanced CO_2_ Capture Using TiO_2_ Nanoparticle-Functionalized Solvent: A Study on Absorption Experiments

**DOI:** 10.3390/nano15050352

**Published:** 2025-02-24

**Authors:** Alice Chillè, Nicola Verdone, Mattia Micciancio, Giorgio Vilardi

**Affiliations:** Department of Chemical Engineering Materials Environment, Sapienza University of Rome, via Eudossiana 18, 00184 Rome, Italy; alice.chille@uniroma1.it (A.C.); nicola.verdone@uniroma1.it (N.V.); mattia.micciancio@uniroma1.it (M.M.)

**Keywords:** CO_2_ capture, reactive absorption, potassium carbonate, TiO_2_ nanoparticles

## Abstract

The growing amount of carbon dioxide (CO_2_) in the atmosphere significantly contributes to global warming and climate change. This study focuses on the use of aqueous potassium carbonate (K_2_CO_3_) solutions as a solvent for CO_2_ absorption, emphasizing the role of titanium dioxide (TiO_2_) nanoparticles in enhancing performance. A detailed understanding of reaction kinetics and the dynamic behavior of the absorber is crucial for optimizing the process. However, critical parameters such as the rate constant k_OH_ of the reaction between CO_2_ and OH^-^ in K_2_CO_3_ solutions are rarely found in existing studies. This work investigates the kinetics of CO_2_ absorption in 25 wt% K_2_CO_3_ solutions at three temperatures (40, 55, and 70 °C), varying concentrations of TiO_2_ nanoparticles to identify optimal conditions. Reaction rates were measured in a stirred cell reactor, and the data were interpreted using Danckwerts theory. The results revealed a notable improvement in absorption efficiency with the addition of nanoparticles, and the study also pinpointed optimal operational parameters to prevent sedimentation issues. The presence of TiO_2_ nanoparticles was found to enhance the solution’s physical properties, such as diffusivity and surface tension, which facilitated an improved mass transfer. The best performance was achieved with a TiO_2_ concentration of 0.06 wt% at 70 °C, leading to an increase of diffusivity value equal to 1.5 times and, as a consequence, the same increase has been observed for the overall reaction rate. In contrast, higher or lower concentrations negatively impacted efficiency due to poor dispersion or nanoparticle agglomeration. These results provide practical insights for developing more efficient and sustainable CO_2_ capture methods, contributing to solutions for the climate crisis.

## 1. Introduction

Carbon dioxide (CO_2_) plays a fundamental role in maintaining the Earth’s thermal balance. However, human activities, such as the extensive use of fossil fuels and industrial processes, have led to a significant rise in atmospheric CO_2_ concentrations. According to Lindsey [1], these levels have now exceeded 400 ppm, far above the pre-industrial value of 280 ppm, with an annual growth rate of approximately 2.5 ppm. This relentless rise is a primary driver of global warming and climate change, leading to severe environmental and socioeconomic consequences. Tackling CO_2_ emissions is therefore critical for achieving the goals outlined in the Paris Agreement and ensuring a sustainable future [2].

Among the proposed strategies, carbon capture and storage (CCS) technologies are regarded as an effective means to reduce CO_2_ emissions. CCS technologies can be divided into three main categories: post-combustion (PCC), oxy-fuel combustion, and pre-combustion [3]. Among these, PCC has gained traction due to its adaptability for retrofitting existing industrial facilities and the availability of the technology at a commercial scale (TRL9) [4]. One of the most well-known and efficient PCC technologies is chemical absorption, which typically using solvents such as monoethanolamine (MEA), to selectively remove CO_2_ from flue gases. MEA and other amine-based solvents are highly effective but face significant drawbacks, including toxicity, high regeneration energy requirements, and issues like corrosion and volatility [5,6].

In recent years, potassium carbonate (K_2_CO_3_) has emerged as a promising alternative solvent [7]. The absorption reaction between CO_2_ and potassium carbonate solution is an exothermic process, where carbonate is converted into bicarbonate during the absorption cycle. The overall reactions occurring inside the absorber follow a series of elementary or parallel steps depending on the pH of the system, as shown in Equations (1) and (2).(1)$${CO}_{2}+H_{2}O+K_{2}{CO}_{3}↔{2KHCO}_{3}\;(non-ionic\;illustration)$$(2)$${CO}_{2}+H_{2}O+{{CO}_{3}}^{2-}↔{{2HCO}_{3}}^{-}\;(inionic\;illustration)$$

The pH significantly influences the reaction between CO_2_ and H_2_O, which remains limited as long as the pH is above 8, the range of interest for commercial applications. Under these basic conditions, the primary mechanism is based on the formation of HCO_3_^−^ through the reaction of CO_2_ with OH^−^, which controls the reaction rate, followed by the rapid conversion of HCO_3_^−^ to CO_3_^2−^, as expressed in Equations (3) and (4).(3)$${CO}_{2}+{OH}^{-}↔{{HCO}_{3}}^{-}\;\left(fast\right)$$(4)$$H{{CO}_{3}}^{-}+{OH}^{-}↔{{CO}_{3}}^{2-}+H_{2}O\;(instanteous)$$

Its low cost, reduced corrosivity, and chemical stability make it suitable for large-scale applications [5,8]. Unlike amines, K_2_CO_3_ is less volatile and less environmentally harmful. However, its slower reaction kinetics [9,10] with CO_2_ pose a challenge, often necessitating larger equipment to achieve comparable removal efficiency [11]. To overcome this limitation, researchers have explored various strategies, including the use of nanotechnology to enhance solvent performance. Cullinane and Rochelle have demonstrated that the addition of piperazine (PZ) as a promoter significantly improves CO_2_ absorption kinetics in K_2_CO_3_ solutions, with optimal concentrations ranging from 20% to 40% by weight [9].

In this study, a 25 wt% K_2_CO_3_ solution was used, in accordance with literature reports which suggest that optimal CO_2_ absorption occurs within a concentration range of 20–40 wt% or maximum molar concentrations up to 4 M K_2_CO_3_ [12]. This concentration was chosen as a compromise between absorption efficiency, stability, and solubility of reaction products. Furthermore, potassium carbonate was selected over sodium carbonate (Na_2_CO_3_) or other alternatives due to its higher solubility, which enhances CO_2_ absorption efficiency and reduces the risk of bicarbonate precipitation. Additionally, K_2_CO_3_ exhibits lower corrosivity, minimizing equipment degradation and ensuring better operational stability.

Nanofluids, a concept introduced by Choi in 1995 [12], are fluids containing nanoparticles dispersed within a base liquid. These systems have demonstrated remarkable properties, such as improved thermal conductivity and enhanced mass transfer rates. Kim et al. [13] were among the first to apply nanofluids in absorption refrigeration systems, where the nanoparticles significantly improved mass transfer efficiency. In the field of CO_2_ capture, nanofluids have shown great potential by enhancing the diffusion of CO_2_ molecules from the gas phase to the liquid phase [14].

The exact mechanism behind the enhancement of CO_2_ absorption by nanofluids is still not entirely clear. However, various studies indicate that this improvement is not driven by a single factor but is more likely the result of multiple mechanisms working together [15]. Among the most discussed are four key mechanisms: the shuttle effect [16], the hydrodynamic effect [14], the bubble disruption effect [17], and Brownian motion [15]. It is worth noting that nanomaterials used in mass transfer applications do not absorb CO_2_ directly. Instead, they enhance the liquid’s capacity to capture CO_2_ by improving its physical and chemical properties [18].

Titanium dioxide (TiO_2_), a widely studied semi-conductor, has proven particularly effective for improving the absorption and regeneration performance of solvents when added as nanoparticle [15]. In its anatase phase, TiO_2_ offers a high surface area and excellent catalytic properties, facilitating more efficient CO_2_ absorption and solvent regeneration. Additionally, the work of Deka et al. [19] showed that the use of TiO_2_ nanoparticles in potassium sarcosinate solvents increased the CO_2_ desorption rate by 128% when compared to the case without nanoparticles., highlighting the potential for catalytic approaches in improving both absorption and regeneration efficiency.

In this context, this study investigates the role of TiO_2_ nanoparticles in enhancing the CO_2_ capture efficiency of K_2_CO_3_ solutions. By combining experimental analysis with an evaluation of kinetic absorption, the research aims to contribute to the development of next-generation CCS technologies. The findings provide insights into achieving higher efficiency while reducing the energy demands for regeneration and environmental footprint of CO_2_ capture systems, aligning with the broader goals of industrial decarbonization and sustainability.

## 2. Materials and Methods

### 2.1. Materials

The reagents potassium carbonate (ACS reagent 99.0%) and titanium (IV) oxide nanopowder (99.9%, anatase phase, ~25 nm particle size) were supplied by Sigma Aldrich^®^ (St. Louis, MO, USA) and were used as received. Gaseous carbon dioxide and nitrogen were purchased at technical grade.

### 2.2. Experimental Apparatus

A stirred cell reactor was used to conduct the absorption tests. A schematic representation of the equipment is given in Figure 1. The experimental setup consisted of two cylinders containing the gases required for the absorption tests: one with carbon dioxide (CO_2_) and the other with nitrogen (N_2_). The gas flows were regulated by two mass flow controllers (MFCs), positioned upstream of a Y-valve that allowed the gases to mix before entering the vessel containing the absorbent. During the tests, gas flow rates were set at 255 mL/min for N_2_ and 45 mL/min for CO_2_ (15 vol%), a ratio chosen to replicate the typical composition of flue gases from a coal-fired power plant.

#### 2.2.1. Absorbent Preparation and Reactor Setup

The base absorbent consisted of a 100 mL solution of 25 wt% K_2_CO_3_. For the preparation of the nano-absorbent, TiO_2_ nanoparticles were dispersed into the base fluid. To ensure homogeneous dispersion and prevent nanoparticle agglomeration, the nanofluid was subjected to ultrasonic treatment (40 kHz) for 20 min before each test. Four concentrations of TiO_2_ nanoparticles were tested (0.01 wt%, 0.06 wt%, 0.1 wt%, and 0.25 wt% of nanoparticles), based on commonly reported values in the literature to optimize absorption in nanofluids and identify the most favorable operating conditions.

The vessel containing the absorbent was equipped with a hermetically sealed lid with inlets for gas feed and outlet and placed on a magnetic stirrer (AREX 5 Digital, Velp Scientifica Srl, Usmate, Italy) set at 300 rpm to ensure uniform mixing of the liquid phase. The system was also fitted with an external water jacket connected to a thermostatic bath (CORIO CD-200F, Julabo, Seelbach, Germany) to maintain a constant temperature. Experiments were conducted at three different temperatures: 40, 55, and 70 °C.

#### 2.2.2. Experimental Procedure and Gas Analysis

The experimental system included a bubble condenser followed by a silica gel dryer to remove moisture from the gas stream before analysis via an infrared analyzer (4400IR ADEV, Cesano Maderno, Italy, measurement range 0–40% and a resolution of 0.01 vol% CO_2_). Condensed water and solvent were recovered in the vessel, minimizing solvent loss during the tests. The IR analyzer allowed real-time monitoring of the amount of CO_2_ not captured by the liquid, providing data on the absorption efficiency of the different tested systems.

The tests were conducted in batch mode for the liquid phase and continuous mode for the gas phase. Before each test, the system was purged with nitrogen gas to remove residual air. With the aid of continuous monitoring, it was verified that CO_2_ levels were undetectable. Absorption efficiency was determined by continuously monitoring the variation in unabsorbed CO_2_ in the gas outlet stream using an IR analyzer. The IR analyzer provided real-time measurements of the CO_2_ concentration in the exiting gas flow, generating a time-dependent profile of CO_2_ levels. This allowed us to track the absorption process dynamically and identify the point at which the test is completed.

At the start of each experiment, the CO_2_ concentration in the outlet stream was zero, indicating that the absorbent was actively capturing the gas. Over time, as the absorbent’s capacity decreased, the measured CO_2_ concentration gradually increased. The absorption reaction was considered complete when the IR analyzer recorded a stable CO_2_ concentration of 15 vol%, matching the inlet concentration of the gas mixture. At this point, no further CO_2_ removal was occurring, indicating that the system had reached saturation.

To evaluate the increase in absorption efficiency, experiments were first conducted on a reference system, using the base K_2_CO_3_ solution as the absorbent, followed by tests with nanofluids obtained by adding small amounts of TiO_2_ nanoparticles to the same solution.

Figure 2 illustrates the block flow diagram of the research workflow, outlining the key stages of the study, from problem identification to data analysis and result interpretation.

### 2.3. Assessment of CO_2_ Absorption Rate

The amount of CO_2_ absorbed was determined by analyzing the data obtained from the IR measurements. The CO_2_ absorption rate was estimated based on the following assumptions: (i) CO_2_ behaves as an ideal gas, (ii) the mass transfer resistance in the gas phase is negligible (iii) the gas–liquid interfacial area coincides with the cross-sectional area of the reactor. Based on these assumptions, the flux of the absorbed gas in solution ($N_{CO_{2}}\;[mol\;s^{-1}]$) can be obtained by Equation (5) [20]:(5)$$N_{CO_{2}}=K_{G}A(P_{CO_{2}}-P_{CO_{2}}^{eq})$$
where $P_{CO_{2}}\;[Pa]$ and $P_{CO_{2}}^{eq}\;$[Pa] is the CO_2_ partial pressure in the cell and its equilibrium pressure, $A\;[m^{2}]$ is the gas–liquid interfacial area and $K_{G}$ is the global mass transfer coefficient. $K_{G}$ is described as the inverse sum of the local mass transfer coefficients in the gas ($k_{G}$) and liquid side ($k_{L}$), modified by Henry’s law constant ($H_{CO_{2}\;}[P\;am^{3}\;{mol}^{-1}]$) for CO_2_ and the enhancement factor (E), as mentioned in Equation (6).(6)$$\frac{1}{K_{G}}=\frac{1}{k_{G}}+\frac{H_{CO_{2}}}{E\cdot k_{L}}$$

This global coefficient expression can be further simplified, as the local mass transfer coefficient in the gas phase is negligible compared to the liquid phase, as shown in Equation (7).(7)$$K_{G}=\frac{k_{L}\cdot E}{H_{CO_{2}}}\;$$

To study the kinetics of CO_2_ with K_2_CO_3_ solutions, it is important that conditions for CO_2_ absorption were chosen to ensure that the process occurs in the “pseudo-first-order reaction regime” [21] given by Equation (8).(8)3<Ha≪E
where the values of Hatta number (Ha) and enhancement factor are calculated by Equations (9) and (10). In carbonate solutions, the pseudo-first-order condition is satisfied by using Danckwerts’ criterion [22]. Under these conditions, E is equal to Ha, and consequently, $K_{G}$ is obtained by Equation (11)(9)$$Ha=\frac{\sqrt{k_{ov}\cdot D_{CO_{2}}}}{k_{L}}\;$$(10)$$E=1+\frac{D_{CO_{2\;}}[solvent]}{{bD}_{CO_{2}}{\left[CO_{2}\right]}_{i}}$$(11)$$K_{G}=\frac{N_{CO_{2}}}{A\left(P_{CO_{2}}-P_{CO_{2}}^{eq}\right)}=\frac{\sqrt{k_{ov}D_{CO_{2}}}}{H_{CO_{2}}}$$
where $k_{ov}\;[s^{-1}]$, $D_{CO_{2}\;}\left[m^{2}s^{-1}\right]$ and b are the overall reaction rate constant, the diffusivity of CO_2_ and the stoichiometric coefficient of solvent, respectively. Using Equation (12) and by multiplying both sides of the equation by the factor $\frac{RT}{V}$, the flux $N_{CO_{2}}\;[Pa\;s^{-1}]$ can be reformulated as follows:(12)$$N_{CO_{2}}=\frac{\sqrt{k_{ov}D_{CO_{2}}}}{H_{CO_{2}}}\frac{RT}{V}A\left(P_{CO_{2}}-{P_{CO}}_{2}^{eq}\right)=m∆P_{CO_{2}\;\;\;}$$
where $m\;[s^{-1}]$ is the slope. According to this equation, m is obtained from the experimental data for each test by plotting $∆P_{CO_{2}}$ absorbed vs $N_{CO_{2}}$.

## 3. Results and Discussion

### 3.1. Kinetic Parameters Estimation

Accurate determination of kinetic parameters for carbon dioxide reactions in potassium carbonate solutions requires precise knowledge of the physical solubility of CO_2_ in the reactive solution. However, direct measurement of CO_2_ solubility is inherently challenging due to experimental limitations. As a result, the solubility of nitrous oxide (N_2_O) is commonly used as an indirect evaluation to estimate the solubility of CO_2_. This approach relies on the established correlation between the solubility of CO_2_ and N_2_O in water and the measured solubility of N_2_O in the target solution, following the principle known as the “N_2_O analogy” [23,24] and expressed by the relationships in Equations (13) and (14).(13)$$\left(\frac{H_{{CO}_{2},K2CO3}}{H_{N_{2}O,K2CO3}}\right)=\left(\frac{H_{{CO}_{2},w}}{H_{N_{2}O,w}}\right)$$(14)$$\left(\frac{D_{{CO}_{2},PC}}{D_{{CO}_{2},w}}\right)={\left(\frac{\mu_{w}}{\mu_{PC}}\right)}^{0.818}$$

Several equations provide a reliable estimation of the solubility and diffusivity parameters for CO_2_ and N_2_O in aqueous solutions, thereby facilitating the modeling of gas absorption processes. According to Versteeg and Van Swaaij [25], based on the available data of solubility and diffusivity of N_2_O and CO_2_ in water, four practical equations were derived, as presented in Equations (15)–(18).(15)$$H_{N_{2}O,w}\;\left[kPam^{3}{kmol}^{-1}\right]=8.5470\;\cdot \;10^{6}e^{\left(\frac{-2284}{T\;\left[K\right]}\right)}\;$$(16)$$H_{{CO}_{2},w}\;\left[kPam^{3}{kmol}^{-1}\right]=2.8249\;\cdot \;10^{6}e^{\left(\frac{-2044}{T\;\left[K\right]}\right)}\;$$(17)$$D_{N_{2}O,w}\;\left[m^{2}s^{-1}\right]=5.07\;\cdot \;10^{-6}e^{\left(\frac{-2371}{T\;\left[K\right]}\right)}$$(18)$$D_{{CO}_{2},w}\;\left[m^{2}s^{-1}\right]=2.35\;\cdot \;10^{-6}e^{\left(\frac{-2119}{T\;\left[K\right]}\right)}$$

To determine the Henry’s constant for N_2_O in water ($H_{N_{2}O,w}$) and for CO_2_ in water ($H_{{CO}_{2},w}$), and subsequently utilize Equation (13) to calculate the Henry’s constant for CO_2_ in a 25 wt% K_2_CO_3_ solution ($H_{{CO}_{2},K2CO3}$), we relied on the experimental data provided by Knuutila et al. [23]. These values were essential to ensure the accuracy and consistency of the calculations under the specified conditions. The data obtained for Henry’s constant from the application of these equations at the three temperatures of the tests performed are shown in Table 1.

The diffusivities of N_2_O in water ($D_{N_{2}O,w}$) and CO_2_ in water ($D_{{CO}_{2},w}$) can be determined using Equations (17) and (18), whereas the diffusivities of CO_2_ in 25 wt% K_2_CO_3_ solution ($D_{{CO}_{2},PC}$) is calculated using the correlation of Joosten and Danckwert [26] derived from a modified Stokes-Einstein equation, as outlined in Equation (19).(19)$$\left(\frac{D_{{CO}_{2},PC}}{D_{{CO}_{2},w}}\right)={\left(\frac{\mu_{w}}{\mu_{PC}}\right)}^{0.818}$$
where $\mu_{w}\;$and $\mu_{PC}$ represent the viscosities of water and 25 wt% K_2_CO_3_ solution, respectively. The value of $\mu_{w}$ was taken from the work of Korson et al. [27] while the value of $\mu_{PC}$ was obtained from the study by Ye and Lu [28]. Using these equations, the results obtained are summarized in Table 2.

### 3.2. Ionization Constants

The stoichiometric constants for the ionization of carbonic acid ($K_{1}$) and bicarbonate ion ($K_{2}$) into K_2_CO_3_ solutions were evaluated based on temperature dependency, as described by Equations (20) and (21) [22]. The results are presented in Table 3.(20)$$\log\left(K_{1}\right)=-\frac{3404.7}{T}+14.843-0.03279T$$(21)$$\log\left(K_{2}\right)=-\frac{2902.4}{T}+6.498-0.0238T$$

### 3.3. Kinetic Constants

The rate constant for the hydration reaction of CO_2_ ($k_{H_{2}O}$) was calculated as a temperature-dependent parameter using the relationship provided in the work of Peirce et al. work [22], as presented in Equation (22).(22)$$\log\left(k_{H_{2}O}\right)=329.85-110.541\log\left[T\right]-\frac{17265.4}{T}\;\;$$

Additionally, the hydroxylation rate constant for CO_2_ ($k_{OH}$) in aqueous potassium carbonate solutions was determined as a function of both temperature and ionic strength (I), based on the Equations (23)–(26), presented in the work of X. Ye and Y. Lu [28]. These correlations are applicable within the temperature range of 25 to 80 °C and for ionic strengths up to 12 kmol·m^−3^.(23)$$\ln\left(k_{OH}\right)=\ln\left(A\right)-\frac{E_{a}}{RT}$$(24)$$E_{a}=47.03$$(25)$$\ln\left(A\right)=0.24I+26.4$$(26)$$I=\frac{1}{2}\sum_{i=1}^{N}C_{i}z_{i}^{2}$$
where $C_{i}$ is the ionic species concentration and $z_{i}$ is the associated ion charge. The experimental data were processed according to the theory of reactive absorption to evaluate the kinetic constants, and the results obtained are reported in Table 4.

As evidenced by these results, the hydration constant is negligible in the calculation of the total reaction kinetics, as the contribution from the hydroxylation reaction at pH > 8 results the limiting stop of the overall CO_2_ conversion rate. Consequently, the equation can be simplified as outlined Equation (27).(27)$$k_{OV}=k_{H20}+k_{OH}\cdot \left[{OH}^{-}\right]{\;\;\;\to \;\;\;k}_{OV}=k_{OH}\cdot \left[{OH}^{-}\right]\;$$

The kinetics of the hydroxylation reaction can be described by the pseudo-first-order kinetic model, as previously mentioned. This approach is justified by the assumption that the concentration of ${OH}^{-}$ in the buffer solution remains constant at the gas–liquid interface during CO_2_ absorption.

### 3.4. Absorption Without TiO_2_

Initial tests were conducted on a reference system, using a base fluid of 25 wt% aqueous solution of K_2_CO_3_ as the absorbent. The data collected, presented in Figure 3, demonstrate that reactive CO_2_ absorption is significantly more efficient at 70 °C compared to lower temperatures. This result can be attributed to several factors related to both reaction kinetics and the physical properties of the fluid.

At higher temperatures, CO_2_ molecules and the fluid gain increased kinetic energy, resulting in more frequent and intense collisions between the gas and the solvent. This phenomenon enhances the likelihood of CO_2_ being absorbed and reacting with the fluid. Although carbon dioxide solubility tends to decrease with increasing temperature, the combined effect of faster reaction rates more than compensates for this reduction. At 70 °C, chemical reactions occur significantly faster, leading to quicker and more effective CO_2_ uptake.

However, it is important to note that excessive temperature increase may lead to undesirable effects, such as fluid degradation. Therefore, optimizing operating conditions requires a careful balance between absorption efficiency and the potential risks associated with higher temperatures.

### 3.5. Absorption with TiO_2_

Subsequent experiments were conducted using nanofluids prepared by dispersing TiO_2_ nanoparticles into the K_2_CO_3_ solution. The first set of tests, with a nanoparticle concentration of 0.01 wt%, demonstrated a significant improvement in CO_2_ absorption compared to the base fluid, as illustrated in Figure 4. This enhancement is consistent with the theoretical framework of the study, which predicts that nanoparticles can enhance mass and heat transfer processes, thereby improving absorption efficiency.

Further tests were conducted with higher nanoparticle concentrations of 0.06 wt% and 0.1 wt%, to evaluate the CO_2_ absorption performance under the same operating conditions. The results of these experiments are presented in Figure 5 and Figure 6.

The analysis of the collected data clearly demonstrates that the addition of nanoparticles to the fluid significantly enhances CO_2_ absorption compared to the base fluid. This confirms the positive influence of nanoparticles on the process efficiency. The observed improvement is primarily attributed to the increase in overall mass diffusivity, driven by the rise in the Schmidt number and the frequency of molecular collisions. Furthermore, the Brownian motion of the nanoparticles contributes to reducing the thickness of the liquid film around gas bubble, thereby accelerating the absorption process [29].

However, it is worth noting that the viscosity and density of the fluid also increase with higher nanoparticle concentrations, which could potentially reduce mass transfer efficiency. Fortunately, this effect becomes significant only at nanoparticle volume fractions exceeding 2% [29].

The optimal temperature was identified as 70 °C, as can be observed from the graph in every test. This behavior is consistent because a temperature of 70 °C provides an ideal balance between reaction kinetics, thermodynamic efficiency, and practical operation. At this temperature, high absorption rates are achieved, energy requirements are minimized, and the equilibrium is maintained to favor bicarbonate formation.

The results of the tests conducted with a nanoparticle concentration of 0.25 wt% are illustrated in Figure 7. During these experiments, partial sedimentation of agglomerated TiO_2_ nanoparticles was observed at the bottom of the vessel. This sedimentation led to a slight reduction in performance, attributed to the excessive nanoparticle content. Consequently, concentrations above 0.25 wt% were not tested because higher concentrations in the base fluid would likely result in an even more pronounced decline in performance.

These results highlight the potential of nanoparticle-functionalized solutions to improve CO_2_ absorption, provided that the concentration of nanoparticles is carefully optimized to avoid agglomeration and maintain the stability of the fluid.

### 3.6. Kinetics Results and Evaluation of CO_2_ Diffusion Coefficient

Based on Equation (12) and the laboratory data, the slope coefficient (m) for each test was determined by plotting the absorbed $∆P_{CO_{2}}$ as a function of the flow rate $N_{CO_{2}}$. By having previously calculated the diffusivity and Henry’s constant, it was possible to determine the rate constant using Equation (28).(28)$$m=\frac{\sqrt{k_{ov}D_{CO_{2}}}}{H_{CO_{2}}}\frac{RT}{V}A$$

To examine the effect of nanoparticles, the rate constant values ($k_{ov}$) were calculated and subsequently held constant in the determination of diffusivity in the presence of nanoparticles. This approach enabled a more accurate assessment of the influence of nanoparticles on the CO_2_ diffusivity in the system. The calculated results for the nanofluids diffusivities are presented in Table 5, which demonstrates that the diffusivity of CO_2_ ($D_{CO_{2}}$) in the presence of nanoparticles is higher compared to the base case.

Nanoparticles enhance mass transfer by facilitating the shuttle effect and reducing the thickness of the liquid film. This is supported by experimental results, which demonstrate a faster absorption compared to the base fluid. Notably, the optimal nanoparticle concentration that significantly improved absorption performance is 0.06 wt% at 70 °C.

To predict the trend of the CO_2_ diffusion coefficient in nanofluids with different temperatures and nanoparticle concentrations, Equation (31) proposed by Nagy et al. [30] was utilized. This equation derived from Frössling equation (Equation (29)) and the modified thermal conductivity expression Equation (30) [31], predicts the increased in the diffusion coefficient.(29)$$Sh=2+0.6\;Re^{0.5}\;Sc^{0.3}$$(30)$$k=k_{f}\left(1+B\;Re^{m}\;Sc^{\frac{1}{3}}\;\varphi \right)$$(31)$$D_{nf}=D_{0}\left(1+B\;Re^{m}\;Sc^{\frac{1}{3}}\;\varphi \right)$$

Equation (31) describes how Brownian motion in nanoparticles can enhance the diffusion coefficient by generating increased velocity gradients in the nanofluid, which in turn accelerates the diffusion rate of the absorbed component. The two unknown parameters, B and m, must be determined by the use of experimental data. For m, a value of 1.7 was adopted based on its analogy to the thermal conductivity equation, consistent with the findings of Prasher et al. [32], who reported that for water-based solutions m is 1.6 ± 15%.

The other parameters include the diffusion coefficient in the nanofluid ($D_{nf}$), the diffusion coefficient of the base fluid ($D_{0}$) and the solid volumetric fraction (φ). Additionally, the Reynolds (Re) and Schmidt numbers (Sc) describe the dynamics of the nanoparticles within the fluid. These dimensionless numbers are influenced by nanoparticle-specific properties, such as the kinematic viscosity of the nanofluid ${(\nu }_{nf})$, the nanoparticle density $\left(\rho_{p}\right)$, and their diameter ${(d}_{p})$, as outlined in Equations (32) and (33).(32)$$Re=\frac{1}{\nu_{nf}}\sqrt{\frac{18k_{B}T}{\pi \rho_{p}d_{p}}}\;$$(33)$$\;Sc=\frac{\nu_{nf}}{D_{nf}}$$

After evaluating the parameter B, a 3D polynomial Equation (34) was derived to describe its behavior across different temperatures and solid volumetric fractions. The coefficients of this equation are provided in Equations (35)–(39).(34)$$B\left(T,\varphi \right)=a_{0}+a_{1}T+a_{2}\varphi +a_{3}T\varphi +a_{4}\varphi^{2}$$(35)$$a_{0}=1.70\cdot 10^{3}$$(36)$$a_{1}=-1.00\cdot 10^{2}$$(37)$$a_{2}=2.79\cdot 10^{5}$$(38)$$a_{3}=4.00\cdot 10^{2}$$(39)$$a_{4}=-3.62\cdot 10^{6}$$

This equation is valid for temperatures in the range of 40–70 °C and solid volumetric fractions between 0.03 and 0.06. The conversion from weight percentage (wt%) of nanoparticles to volume fraction can be obtained by Equations (40) and (41). Nanofluid density ($\rho_{nf}$) is estimated according to the mixture rule [33], where $\rho_{np}$ denotes the density of nanoparticles, while $\rho_{bf}\;$refers to the density of the base fluid.(40)$$\phi =\frac{\rho_{bf}\phi_{m}}{\rho_{np}\left(1-\phi_{m}\right)+\phi_{m}\rho_{bf}}$$(41)$$\rho_{nf}=\rho_{bf}\left(1-\phi \right)+\phi \rho_{np}$$

In this equation $\phi_{m}\;and\;\phi \;$are the mass fraction and volume fraction of nanoparticles, respectively.

The diffusion coefficient data corresponding to a mass fraction of 0.25 wt% were excluded from the evaluation of parameter B due to the observed deposition of part of nanoparticles during experimental tests, which compromised the stability of the nanofluid. Figure 8 illustrates the trend of the parameter B as a function of temperature (T) and solid volumetric fraction (φ).

The figure shows a predominantly linear decrease in B with increasing temperature, while changes in solid volumetric fraction produce nonlinear effects, particularly at higher values of φ. This trend aligns with physical behavior of the nanofluid because when temperatures rise Brownian motions get stronger and higher nanoparticle concentrations increase inter-particle interactions, influencing the overall diffusion dynamics.

To evaluate the effectiveness of CO_2_ absorption achieved in this study, a comparative analysis was conducted with data reported in the literature for various nanomaterials. Table 6 presents a comparison of different base absorbents, nanoparticles used, operating conditions, and the enhancement ratio (E_R_) of CO_2_ removal efficiency. The E_R_ represents the improvement in CO_2_ absorption efficiency when nanoparticles are added to the base absorbent. It is calculated as the ratio between the absorption rate in the nanofluid system and that in the base absorbent without nanoparticles. An E_R_ value greater than 1 indicates a positive effect of the nanoparticles on the absorption process.

## 4. Conclusions

In this work, the impact of nanofluids on CO_2_ capture was studied using a stirred cell reactor. The research focused on the absorption of CO_2_ in a 25 wt% K_2_CO_3_ solution enhanced with a small amount of TiO_2_ nanoparticles to optimize the carbon dioxide capture efficiency.

The experimental results revealed that a nanoparticle concentration of 0.06 wt.% significantly improved absorption performance. However, deviations from this optimal concentration negatively affected efficiency: lower concentrations led to inadequate nanoparticle dispersion, while higher concentrations caused agglomeration, hindering the nanofluid’s CO_2_ capture capabilities.

Despite these promising results, the study has limitations, including the exclusive use of a 25 wt% K_2_CO_3_ solution, a baseline absorbent with relatively low absorption capacity compared to other alternatives with potentially higher absorption. Additionally, the study focused solely on the absorption phase, omitting the desorption process, which is a critical component for evaluating the full efficiency of carbon capture technologies. The long-term stability of the nanoparticle-functionalized solution was not assessed, leaving uncertainties regarding potential degradation or changes in effectiveness over extended operational periods. Furthermore, the experiments were conducted in a controlled laboratory setting, which may not fully capture the complexities and challenges of industrial-scale applications.

In this study, a baseline absorbent with relatively low absorption capacity was used. Future research could focus on exploring more efficient absorbents than the K_2_CO_3_ solution and optimizing nanoparticle design. Additionally, since desorption is a critical step in the carbon capture process, efforts should focus on enhancing its efficiency through the development of advanced techniques that facilitate faster CO_2_ release with lower energy consumption. Building on these findings, future research should focus on advanced kinetic modeling and scaling the technology for industrial applications, aiming to bridge the gap between laboratory results and practical implementation.

## Figures and Tables

**Figure 1 nanomaterials-15-00352-f001:**
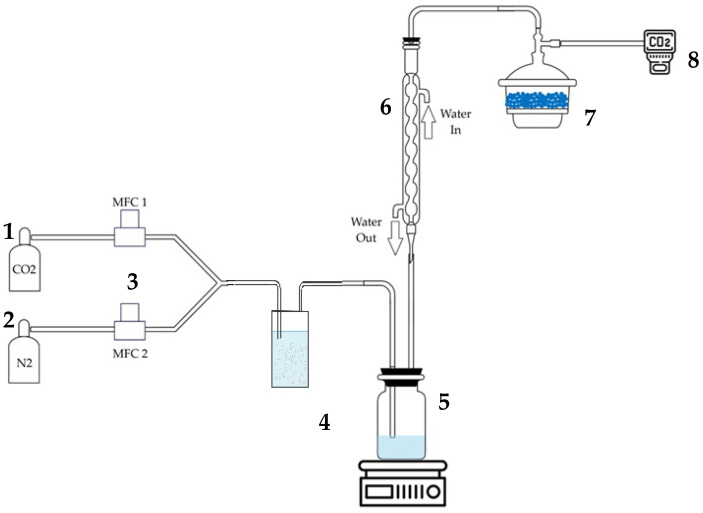
Experimental apparatus for CO_2_ absorption tests: (1) carbon dioxide cylinder, (2) nitrogen cylinder, (3) mass flow controllers, (4) reactor for mixing the simulated gas, (5) jacketed stirred cell reactor, (6) bubble condenser, (7) silica gel dryer, (8) IR analyzer (DAQ: data acquisition unit).

**Figure 2 nanomaterials-15-00352-f002:**
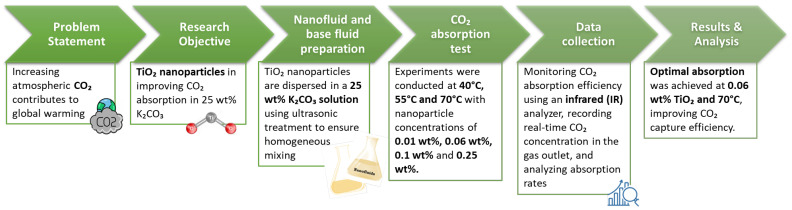
Block flow diagram.

**Figure 3 nanomaterials-15-00352-f003:**
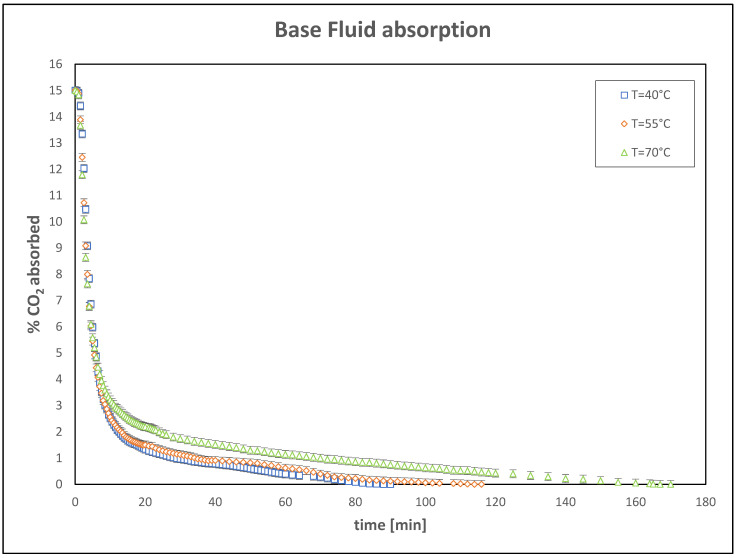
Trend of CO_2_ absorbed in the base fluid.

**Figure 4 nanomaterials-15-00352-f004:**
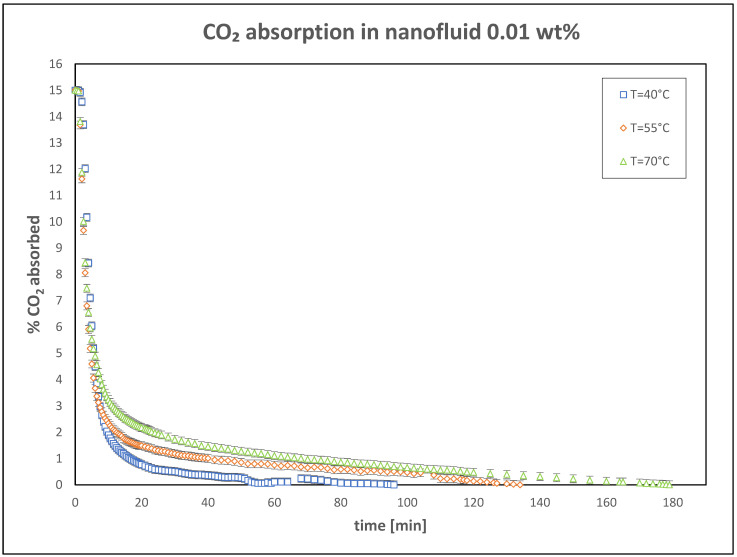
Trend of CO_2_ absorbed in nanofluid: 0.01 wt%.

**Figure 5 nanomaterials-15-00352-f005:**
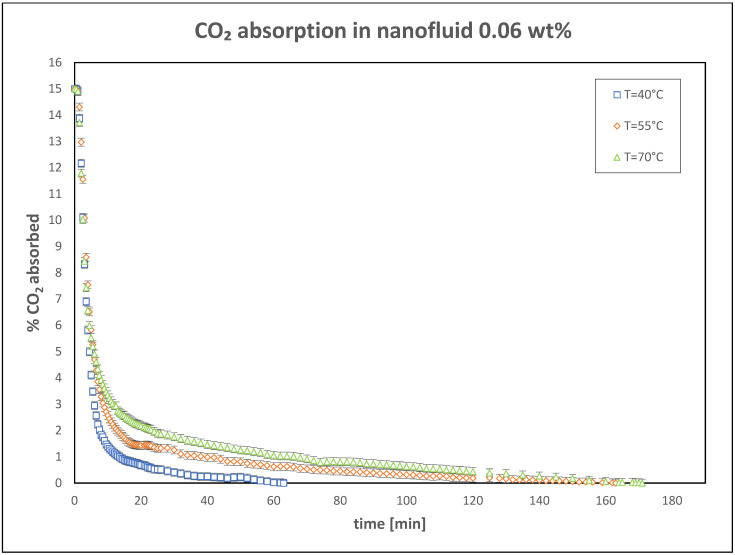
Trend of CO_2_ absorbed in nanofluid: 0.06 wt%.

**Figure 6 nanomaterials-15-00352-f006:**
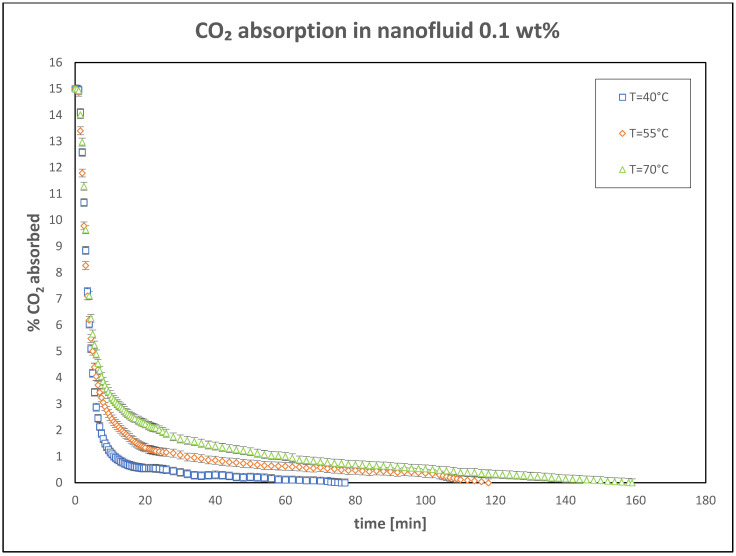
Trend of CO_2_ absorbed in nanofluid: 0.1 wt%.

**Figure 7 nanomaterials-15-00352-f007:**
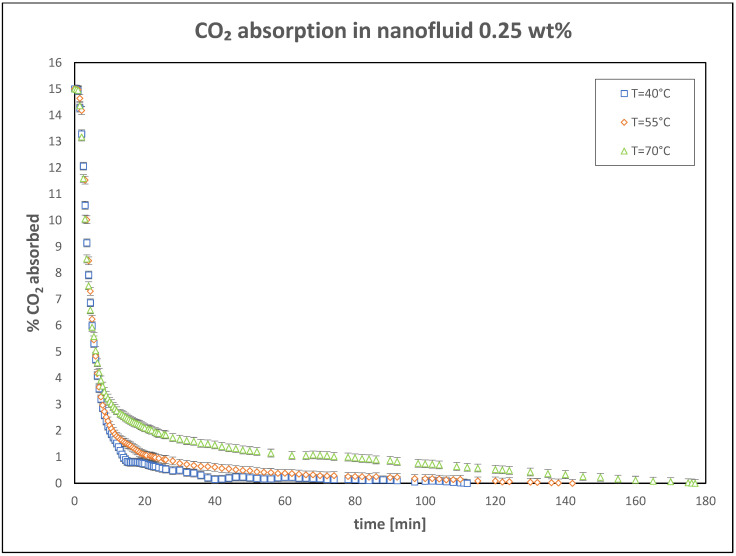
Trend of CO_2_ absorbed in nanofluid: 0.25 wt%.

**Figure 8 nanomaterials-15-00352-f008:**
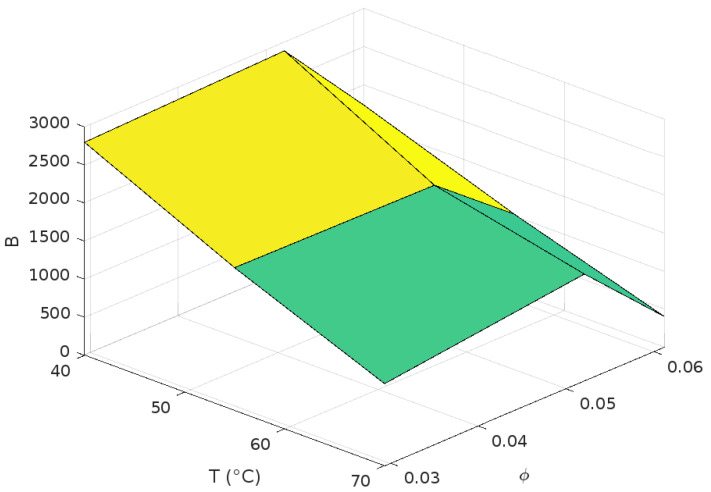
Three-dimensional surface plot where the dependent variable *B* is plotted against two independent variables: temperature (*T*) in degrees Celsius and the volumetric fraction of nanoparticles (φ) in a nanofluid.

**Table 1 nanomaterials-15-00352-t001:** Values of Henry’s constant in the 25% K_2_CO_3_ solution.

T [K]	H_N2O,w_ [Pa m^3^ mol^−1^]	H_CO2,w_ [Pa m^3^ mol^−1^]	H_CO2,K2CO3_ [Pa m^3^ mol^−1^]
313.15	5810.73	4133.01	21,615.91
328.15	8110.05	5569.86	25,299.96
343.15	10,994.05	7312.97	28,792.79

**Table 2 nanomaterials-15-00352-t002:** Diffusivity values in the 25 wt% K_2_CO_3_ solution.

T [K]	μ_CO2,W_ [Cp] [27]	μ_CO2,K2CO3_ [Cp] [23]	D_CO2,W_ [m^2^ s^−1^]	D_CO2,K2CO3_ [m^2^ s^−1^]
313.15	0.653	1.341	2.71 × 10^−9^	1.50 × 10^−9^
328.15	0.504	1.089	3.69 × 10^−9^	1.96 × 10^−9^
143.15	0.404	0.867	4.89 × 10^−9^	2.62 × 10^−9^

**Table 3 nanomaterials-15-00352-t003:** Stoichiometric constants for the ionization of carbonic acid ($K_{1}$) and bicarbonate ion ($K_{2}$) in K_2_CO_3_ solutions.

T [K]	log (K_1_)	K_1_	log (K_2_)	K_2_
313.15	−6.298	5.039 × 10^−7^	−10.223	5.979 × 10^−11^
328.15	−6.292	5.100 × 10^−7^	−10.157	6.971 × 10^−11^
343.15	−6.331	4.669 × 10^−7^	−10.127	7.463 × 10^−11^

**Table 4 nanomaterials-15-00352-t004:** Rate constants of $k_{OH}$ and $k_{H_{2}O}.$

T [K]	k_OH_ [m^3^ kmol^−1^ s^−1^]	k_H2O_ [m^3^ kmol^−1^ s^−1^]
313.15	29,468.97	0.068
328.15	67,283.31	0.128
343.15	142,951.59	0.182

**Table 5 nanomaterials-15-00352-t005:** Calculation of Nanofluid Diffusivity.

Nanofluid [wt%]	T [°C]	D_CO2_ (×10^−9^) [m^2^ s^−1^]
0	40	1.56
	55	1.96
	70	2.61
0.01	40	2.42
	55	2.76
	70	3.22
0.06	40	3.12
	55	2.09
	70	3.98
0.1	40	2.72
	55	2.92
	70	3.13
0.25	40	1.62
	55	2.38
	70	2.53

**Table 6 nanomaterials-15-00352-t006:** Comparison of CO_2_ absorption efficiency for different nanomaterials reported in the literature.

Base Absorbent	Nanoparticles	Temperature [°C]	Concentration	E_R_	Ref.
MEA	SiO_2_	103	0.12 wt%	>1.1	[34]
	TiO_2_	103	0.12 wt%	>1.1	
	Al_2_O_3_	103	0.12 wt%	>1.1	
MDEA	TiO_2_	37	0.09 wt%	1.25–1.65	[35]
MDEA	CNT	23	0.02 wt%	1.35	[36]
H_2_O	SiO_2_	25	0.1 wt%	1.07	[37]
	ZnO	25	0.1 wt%	1.14	
H_2_O	Al_2_O_3_	25	0.1 wt%	1.21	[38]
	SiO_2_	25	0.01 wt%	1.45	
	Fe_2_O_3_	25	1 wt%	1.16	
MeOH	SiO_2_	17	0.05 vol%	1.011	[39]
	Al_2_O_3_	17	0.05 vol%	1.012	
	TiO_2_	17	0.05 vol%	1.046	
MeOH	Al_2_O_3_	21	0.1vol%	1.27	[40]
K_2_CO_3_	TiO_2_	70	0.06 wt%	1.52	This work

## Data Availability

The data presented in this study are available on request from the corresponding author due to protect the intellectual properties.

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
