# Peer review of "Enhanced CO2 Capture Using TiO2 Nanoparticle-Functionalized Solvent: A Study on Absorption Experiments"

_nanomaterials, 2025, doi:10.3390/nano15050352_

Round 1
Reviewer 1 Report
Comments and Suggestions for Authors
Chille et al. have aimed to investigate the role of TiO2 nanoparticles in enhancing the CO2 absorption efficiency of K2CO3 solutions for possible development of next-generation CCS technologies to reduce energy demands for regeneration and environmental footprint of CO2 capture systems, thereby aligning with the broader goals of industrial decarbonization and sustainability. Although the findings of this study are of interest in the field, the following issues need to be addressed for a possible publication in Nanomaterials:
- Some key quantitative data on optimized conditions, carbon dioxide absorption efficiency and reaction rate should be specified in the abstract.
- Separate keywords “TiO2” and “nanoparticles” should be combined as one keyword “TiO2 nanoparticles”.
- Schematic diagram illustrating the workflow and key outcomes should be included.
- Description of all the abbreviations and variables used in Tables and Figures should be included in the respective table footnotes and figures captions.
- Be consistent with the usage of wt% or %wt throughout the text, tables and figures.
- The %wt wherever it is should be accompanied by the name of the variable in %wt.
- Section 2.2 should be split into several subheadings.
- Section 2 should also end with data analysis or statistical analysis description done in this study.
- Figure 1 caption should briefly explain the experimental set-up.
- In Table 2, the unit of K1 and K2 should be provided.
- Figures 2, 3, 4, 5 and 6 should be reduced in size and combined as parts of one figure for easy comparison.
- In Table 5, “%wt” should be “Nanofluid (%wt)”.
- The caption of Figure 7 should be elaborated with more description.
- All the equations and units in the manuscript should be double-checked for correctness.
- A new table should be included for comparing the absorption efficiency of CO2 obtained in this study with those reported in the literature for other nanomaterials.
- The limitation of this study should be highlighted in the conclusion. Also, the conclusion should be of one paragraph.
Author Response
We sincerely appreciate your insightful comments and constructive suggestions, which have been invaluable in improving the quality and clarity of our manuscript. Below, we provide a detailed point-by-point response to each of your recommendations:
Comment 1:Some key quantitative data on optimized conditions, carbon dioxide absorption efficiency and reaction rate should be specified in the abstract
Response 1:We have incorporated key quantitative data regarding optimized conditions, carbon dioxide absorption efficiency, and reaction rate into the abstract.
Comment 2: Keywords “TiO2” and “nanoparticles” should be combined as one keyword “TiO2 nanoparticles”.
Response 2: The keywords “TiO₂” and “nanoparticles” have been merged into a single term: “TiO₂ nanoparticles.”
Comment 3: Schematic diagram illustrating the workflow and key outcomes should be included.
Response 3: A schematic diagram has been added to illustrate the workflow and key outcomes of the study.
Comment 4: Description of all the abbreviations and variables used in Tables and Figures should be included in the respective table footnotes and figures captions.
Response 4:Descriptions of all abbreviations and variables have been included to ensure clarity.
Comment 5: Be consistent with the usage of wt% or %wt throughout the text, tables and figures.
Response 5: The usage of "wt%" has been standardized throughout the manuscript, including the text, tables, and figures.
Comment 6: The %wt wherever it is should be accompanied by the name of the variable in %wt.
Response 6: Wherever "wt%" appears, the corresponding variable has been explicitly specified.
Comment 7: Section 2.2 should be split into several subheadings.
Response 7: Section 2.2 has been divided into multiple subheadings to improve structure and readability.
Comment 8: Section 2 should also end with data analysis or statistical analysis description done in this study.
Response 8: A description of data processing method employed in this study has been added at the end of Section 2.
Comment 9: Figure 1 caption should briefly explain the experimental set-up.
Response 9:The caption of Figure 1 has been revised to provide a more comprehensive explanation of the experimental setup.
Comment 10: In Table 2, the unit of K1 and K2 should be provided.
Response 10: The variables K₁ and K₂ are dimensionless; therefore, no units have been provided in Table 2.
Comment 11: Figures 2, 3, 4, 5 and 6 should be reduced in size and combined as parts of one figure for easy comparison.
Response 11:The figures have not been merged, as doing so would have compromised data clarity and readability due to overlapping elements. Instead, we have ensured that the figures remain clearly distinguishable while maintaining their interpretability.
Comment 12: In Table 5, “%wt” should be “Nanofluid (%wt)”.
Response 12: The label “%wt” in Table 5 has been revised to “Nanofluid (%wt)” for clarity.
Comment 13: The caption of Figure 7 should be elaborated with more description.
Response 13: The caption of Figure 7 has been expanded with a more detailed description of its content.
Comment 14: All the equations and units in the manuscript should be double-checked for correctness.
Response 14: All equations and units throughout the manuscript have been reviewed and verified for accuracy.
Comment 15: A new table should be included for comparing the absorption efficiency of CO2 obtained in this study with those reported in the literature for other nanomaterials.
Response 15: A new table has been added to compare the CO₂ absorption efficiency obtained in this study with values reported in the literature for other nanomaterials.
Comment 16: The limitation of this study should be highlighted in the conclusion. Also, the conclusion should be of one paragraph.
Response 16:The conclusion has been revised into a single paragraph and now includes a discussion of the study’s limitations, as recommended.
Reviewer 2 Report
Comments and Suggestions for Authors
The paper deals with one of the hot topics, namely fixing CO2 with K2CO3 solution aided with TiO2 nanoparticles. The Authors tested different nanoparticle concentrations and temperatures.
The Authors performed their studies in a well-planned manner. They have studied the CO2 absorption in nanoparticle-free solutions at three different temperatures (40, 55, and 70 °C) and at four different nanoparticle concentrations (0.01, 0.06, 0.1, and 0.25 wt%) at the same temperatures. They have found that the best temperature is 70 °C and the best nanoparticle concentration is 0.06 wt%.
The Reviewer has the following suggestion to improve the manuscript further:
- On page 3, the Authors mentioned that the amount of absorbed CO2 was determined based on IR studies. I suggest explaining the method shortly or citing a paper in which it was described.
- On page 2 the Authors mentioned the K2CO3 solution as a promising solvent. It would be good to show the chemical equation as in the 10.1021/acs.iecr.9b05498 and explain why they decided to use the 25 % solution. Is there any role in using the K-salt instead the cheaper sodium carbonate/at this concentration, e.g. due to the solubility of the products of CO2 absorption?
- On page 3, in the case of Figure 1 it should be good the content of the two solutions. for better understanding. A
- In the case of the references, they are not formatted as it is required by the journal. It must be revised.
- CO2 was taken as an ideal gas, however, CO2 is far enough from ideal gas behaviour, so I suggest using fugacity (fugacity coefficient with pressure values) in eq. 1.
Based on the above-mentioned points the Reviewer suggests a minor revision for the paper. The Authors should do a careful revision of the grammar, formatting, and editing of the manuscript.
Author Response
We sincerely appreciate your valuable comments and constructive suggestions, which have helped us further improve the quality and clarity of our manuscript. Below, we provide a detailed point-by-point response to each of your recommendations:
-
Explanation of the CO₂ absorption method (Page 3)
We have now provided a brief explanation of the method used to determine the amount of absorbed CO₂ based on IR studies. -
K₂CO₃ solution as a solvent (Page 2)
We have included the chemical equation and explained our choice of a 25% K₂CO₃ solution. Furthermore, we have clarified the rationale for using potassium carbonate instead of sodium carbonate, particularly at this concentration, where solubility considerations play a significant role in the CO₂ absorption process. -
Clarification of Figure 1 (Page 3)
We have added information in Figure 1 to enhance clarity and understanding. -
Treatment of CO₂ as an ideal gas
We acknowledge your concern regarding the assumption of CO₂ as an ideal gas. However, under the conditions of our study, CO₂ can be approximated as an ideal gas. To validate this assumption, we conducted a simulation using Aspen Plus, which confirmed the accuracy of the simplification employed. Therefore, we have maintained this approach in our manuscript.
Round 2
Reviewer 1 Report
Comments and Suggestions for Authors
The authors have satisfactorily addressed all the comments raised by reviewers and substantially improved the overall quality of the article. Therefore, I recommend accepting this article for publication in Nanomaterials.